# Health Status in Long-Term Survivors of Hepatoblastoma

**DOI:** 10.3390/cancers11111777

**Published:** 2019-11-11

**Authors:** Bożenna Dembowska-Bagińska, Jolanta Więckowska, Agnieszka Brożyna, Ewa Święszkowska, Hor Ismail, Dorota Broniszczak-Czyszek, Marek Stefanowicz, Wiesława Grajkowska, Piotr Kaliciński

**Affiliations:** 1Department of Oncology, Children’s Memorial Health Institute, 04-730 Warsaw, Polandj.wieckowska@ipczd.pl (J.W.); a.brozyna@ipczd.pl (A.B.); e.swieszkowska@ipczd.pl (E.Ś.); 2Department of Pediatric Surgery & Organ Transplantation; Children’s Memorial Health Institute, 04-730 Warsaw, Poland; h.ismail@ipczd.pl (H.I.); d.broniszczak@ipczd.pl (D.B.-C.); m.stefanowicz@ipczd.pl (M.S.); 3Department of Pathology; Children’s Memorial Health Institute, 04-730 Warsaw, Poland; w.grajkowska@ipczd.pl

**Keywords:** hepatoblastoma, survivors, children, follow-up, long-term health status

## Abstract

The aim of this study was to evaluate the health status of children cured from hepatoblastoma. Forty-five patients with hepatoblastoma treated between 1996–2014 were assessed. The recorded data included sex, age at diagnosis, disease stage, treatment methods, time since diagnosis, and the evaluation of health status domains which included performance status, growth development, hearing, cardiovascular, skeletal, gastrointestinal, genitourinary, neurological, and hematological function. There were 30 boys and 15 girls. The age at diagnosis ranged from one month to 14 years (median one year). At the time of the health status evaluation, the youngest patient was 5.5 years old and the oldest was 21 years of age (median—10 years). All patients were treated according to the Childhood Liver Tumors Strategy Group—SIOPEL recommendations, though they were not active participants of the studies. The median cumulative dose of cisplatin was 520 mg/m^2^ and 360 mg/m^2^ for doxorubicin. Thirty-six patients underwent partial hepatectomy, and nine total hepatectomy and liver transplantation. At a median of nine years from diagnosis, 68% of hepatoblastoma survivors had experienced at least one chronic health condition of any grade. The most frequent late complication was ototoxicity (28.8%), and the most serious were second malignancies (6.6%) and cardiomyopathy (4.4%). Conclusion: Survivors of hepatoblastoma are at risk for long-term complications. They require long-term monitoring for late effects.

## 1. Introduction

Hepatoblastoma is the most common primary liver tumor in children. Over the past three decades, randomized controlled trials for children with hepatoblastoma have demonstrated that chemotherapy consisting of cisplatin with or without doxorubicin administered before and after tumor resection results in overall survival of over 80% [1,2,3]. Although hepatoblastoma therapy is considered less aggressive than treatment for other pediatric malignancies, still hepatoblastoma survivors are at risk of developing long-term treatment-related complications. In the accessible up to date medical literature, there are several publications devoted specifically to the late-effects in children cured of hepatoblastoma [4]. This study aims to assess the long-term health status of these survivors.

## 2. Results

Among 88 children with hepatoblastoma treated at our institute since 1996, 45 who were at least five years from diagnosis were available for analysis. There were 30 boys and 15 girls (Male:Female = 2:1). Among 45 patients, 7 (15.5%) were prematurely born. Their gestational age ranged from 28 to 36weeks (median—34 weeks), weight ranged from 580 to 3080 grams (median—2060 grams). Out of seven premature babies, one was below the 3rd percentile for weight. The rest were over the 50th percentile. In the whole group, 31 of the 45 (68.8%) patients’ weight at birth was over the 50th percentile while at hepatoblastoma diagnosis, 18/45 (40%) had a body weight over the 50th percentile. At the time of treatment, 15.5% of patients (all boys) were below the 10th percentile for height. The age at diagnosis ranged from one month to 14 years (median—one year).

Five patients (11%) presented with lung metastases at diagnosis. There were four patients with overgrowth syndromes, two with Beckwith–Wiedemann Syndrome (BWS), one with Simpson–Golabi–Behmel syndrome, one with hemihypertrophy, and two with Familial Adenomatous Polyposis (FAP).

All but one patient received treatment according to the Childhood Liver Tumors Strategy Group—SIOPEL recommendations from studies running at the time of diagnosis (with neoadjuvant and adjuvant chemotherapy consisting of cisplatin therapy 80–100 mg/m^2^/cycle with or without doxorubicin 60 mg/^2^). The cumulative total dose of cisplatin ranged from 250 to 820 mg/m^2^ (median—520 mg/m^2^) and from 40 to 420 mg/m^2^ (median—360 mg/m^2^) for doxorubicin. Cisplatin was administered as a 24-h infusion in all patients. Magnesium was supplemented during hydration. Mannitol was administered according to protocol recommendations. None of the patients received thiosulphate. One patient received dexrazoxane before the infusion of doxorubicin.

Thirty-six (80%) patients underwent partial hepatectomy. In nine children (three with POSTEXT III and six with POSTEXT IV), total hepatectomy and liver transplantation was performed, one from a deceased donor, and eight from living donors. The overall survival at three years for all patients treated since 1996 was 88.8%.

### 2.1. Health Status and Performance Evaluation

#### 2.1.1. Physical Performance

Six patients were assessed with the Karnofsky and 39 with the Lansky scale. None of them suffered from functional impairment nor activity limitations. They could do normal activities without physical problems. The Karnofsky/Lansky score was 100% in all. 

#### 2.1.2. Physical Development

Twenty percent of males and 5% of females were below the 10th percentile for height (median 25–50 percentile for both sexes) (Figure 1), More than half of the patients were underweight, 64% of males and 50% of females had a body mass index (BMI) below 18.5 percentile (Figure 2). Among the 45 patients 17 had normal body weight at the time of birth (including two premature babies with BMI >90 percentile) and maintained their normal weight. Thirteen children with BMI <25 percentile at the time of the study were underweight at birth and hepatoblastoma diagnosis (including two premature babies, 1 below the third percentile). Nine patients with BMI below 18.5 percentile at the time of the follow-up had normal body weight at birth and hepatoblastoma diagnosis. In five patients (including three premature babies) an increase in BMI was observed from the time of birth to the last follow-up. 

#### 2.1.3. Puberty

All patients had normal stages of puberty described by the Tanner scale.

#### 2.1.4. Hearing

Thirteen patients (28.8%) demonstrated sensorineural hearing impairment, of which six (46%) had severe (Brock 3/4) and seven had mild to moderate (Brock 1/2) hearing loss. At treatment, all of them were below three years of age. Seven (53.8%) have hearing aids. The cisplatin cumulative dose for patients with severe hearing impairment ranged from 300 to 820 mg/m2 (median—480 mg/m^2^) and for those with Brock 1/2 from 300 to 640 mg/m2 (median—480 mg/m^2^).

#### 2.1.5. Cardiovascular Status

Among 40 patients who received anthracyclines, two patients (4.4%) experienced heart insufficiency. One boy with advanced hepatoblastoma (PRETEXT IV) at nine years of age and seven years from hepatoblastoma treatment developed cardiomyopathy managed by cardiologists. The second patient developed cardiomyopathy, 10 years from hepatoblastoma diagnosis, underwent heart transplantation at the age of 20 and died four years later of heart failure. They received cumulative anthracyclines doses of 360 and 420 mg/m^2^_,_ respectively. The rest of the patients had no cardiologic clinical signs nor abnormal echocardiograms. Two patients had hypertension treated with angiotensin-converting-enzyme inhibitor (ACE) and a calcium channel blocker.

#### 2.1.6. Skeletal System 

Three patients had skeletal problems. Two of them experienced bone fractures as a result of minor trauma. It was the first symptom of low bone density. One patient has idiopathic scoliosis with low bone density. Twelve patients who were screened for osteoporosis had normal bone density. 

#### 2.1.7. Nephrotoxicity 

One patient was diagnosed with cisplatin-induced tubular damage leading to magnesia and chronic hypomagnesemia

#### 2.1.8. Genito-Urinary 

One patient, an 11-year-old boy suffers from nocturnal enuresis 

#### 2.1.9. Gastrointestinal

One patient suffers from gastroesophageal reflux disease. Two other patients are under surveillance for familial adenomatous polyposis (FAP), one after treatment of colon cancer, the other with a family history of FAP.

#### 2.1.10. Neurological Disorders

Two patients have epilepsy controlled well with anti-epileptic medication. Epilepsy was diagnosed after hepatoblastoma treatment.

#### 2.1.11. Hematological Disorders

One patient was diagnosed with acute lymphoblastic leukemia (ALL). Another patient, eight years after the third liver transplantation, developed severe autoimmune thrombocytopenia and died due to its complications. There were no other hematological late adverse events. 

#### 2.1.12. Late complications of surgical tumor resection

Two patients developed biliary complications after tumor resection. In one case biliary fistula after extended liver resection was treated with hepaticojejunostomy. No further problems were observed during 11 years follow-up. Another patient developed biliary cirrhosis four years from the primary extended liver resection of large volume hepatoblastoma performed in the first month of age and underwent liver transplantation 

#### 2.1.13. Late Complications after Liver Transplantation

One patient after transplantation of partial graft from his father developed super acute humoral rejection, and urgent retransplantation was performed two days later. 1.5 years after transplantation, he developed thrombocytopenia which was treated successfully with steroid pulses and rituximab. He also underwent single percutaneous balloon dilatation of portal vein anastomosis. At present, he is doing very well three years following transplantation. 

Another patient underwent transplantation in the first year of life and developed biliary complications, which led to the retransplantation two years later, and the 3rd transplantation within next one month due to primary poor function of the graft. During the next eight years, he underwent several percutaneous transvascular balloon dilatations and stenting of the hepatic vein to inferior caval vein anastomosis. He died of complications of acquired idiopathic thrombocytopenia. 

Third patient presented late biliary anastomotic stenosis which was treated successfully with percutaneous transhepatic balloon dilatation. 

The fourth patient developed hypersplenism due to portal vein thrombosis, which was treated successfully with partial splenic embolization (Table 1).

#### 2.1.14. School performance

All patients attended a regular school. One patient with Simpson–Golabi–Bemel syndrome had a history of speech delay. All of the survivors reported satisfactory school achievements in terms of satisfactory report cards and successfully completing classes.

#### 2.1.15. Second malignant neoplasms

Three patients (6.6%) were diagnosed with second malignant neoplasms. One girl at the age of 2.5 years developed acute lymphoblastic leukemia (ALL), two years from hepatoblastoma diagnosis. She was treated according to the Berlin–Frankfurt–Munster (BFM) ALL protocol. Her leukemia treatment was well tolerated. She is alive, disease-free, six years from ALL diagnosis. 

Another girl developed medulloblastoma seven years from hepatoblastoma diagnosis. Despite all methods of persuasion, parents refused irradiation and chemotherapy for her brain tumor. She died from the disease one year from diagnosis of second malignant neoplasm. 

The third patient was diagnosed with colon carcinoma nine years from hepatoblastoma diagnosis. He had no family history of colon cancer nor history suggestive of a cancer family syndrome. Identification of a germ line mutation confirmed familial adenomatous polyposis (FAP). He is alive disease-free, two years from colon cancer diagnosis and treatment with hemicolectomy and chemotherapy.

#### 2.1.16. Deaths

There were two deaths unrelated to hepatoblastoma itself but from the late effects of treatment.

One patient died at the age of 24 of cardiac insufficiency 21 years from hepatoblastoma diagnosis. When he was 13 years old, he developed anthracycline-induced cardiomyopathy after receiving a cumulative dose of doxorubicin of 420 mg/m^2^, and seven years later, he underwent heart transplantation. He did not comply with the recommended treatment rules, stopped taking his immunosuppressive medications, and suffered from the rejection of the transplanted organ. 

The second patient died 10 years after the first liver transplantation. It was a boy who received three liver grafts altogether. He died despite of good third graft function from complications of acute uncontrolled thrombocytopenia of unknown origin, gastrointestinal, and respiratory tract bleeding. One patient described above died of second malignancy. 

Out of 45 long-term survivors of hepatoblastoma, one third had two or more chronic health conditions to a person (Figure 3). 

## 3. Discussion

Advances in the treatment of childhood cancer have led to a growing number of survivors at risk for long-term health complications. Most of the adverse health outcomes are attributed to treatment. Underlying genetic risk factors may also be involved, though they have not yet been well documented [5,6].

While hepatoblastoma has a high survival rate with a combination of conventional complex treatment and transplantation in selected cases, studies have demonstrated that survivors suffer from chronic health conditions of which ototoxicity and cardiotoxicity are the most frequent [4]. These adverse health outcomes are caused by cisplatin and doxorubicin, which are essential components of hepatoblastoma chemotherapy protocols.

The prevalence of cisplatin-induced ototoxicity varies between different regimens and tumor types. More than half of children treated with cisplatin develop permanent hearing impairment, of which about 25% is severe. This also affects hepatoblastoma survivors [7,8,9,10]. We have confirmed this observation in our series where over 28% of children had hearing loss. The risk of hearing impairment is associated with a cumulative dose of cisplatin and a young age at treatment [11]. In our study, children with severe hearing impairment were all below three years of age at treatment and received cisplatin in a dose equal or greater than 300 mg/m^2^. This deficit can progress over time and can occur late [7]. Brock et al. in SIOPEL 6, phase 3 trial investigated the role of sodium thiosulfate in reducing the incidence and severity of hearing loss caused by cisplatin in children with localized hepatoblastoma. Out of 109 randomized patients, 57 received cisplatin followed by sodium thiosulphate and 52 cisplatin alone. The results of this study have clearly shown that sodium thiosulfate given after cisplatin infusion reduces by half the incidence of hearing loss without jeopardizing survival [11]. This could lower the burden of late effects in patients treated of hepatoblastoma in the future, though in a recent Pediatric Hepatic International Tumor Trial (PHITT) [12], the use of sodium thiosulfate is not foreseen. Cisplatin-induced hearing impairment mostly involves a high frequency of sounds, which are significant in communication and educational achievement. Gurney et al. and Bess et al. [13,14] investigated school performance and general learning ability of neuroblastoma and other solid tumor survivors who had treatment-induced hearing loss and found that the hearing deficit is strongly associated with learning problems and special educational needs. In our study, we did not observe this since all our hepatoblastoma survivors with hearing impairment were performing well in educational skills. Our observation might be explained by the small number of patients studied as compared to Bess’s paper where school performance was evaluated in 1218 children. Of importance is an observation from our study that our hepatoblastoma teachers were informed of their special hearing needs. 

Cardiovascular diseases after cancer relapse and second malignant neoplasms (SMN) are the leading cause of morbidity and mortality in survivors of childhood cancer and are largely related to therapy with anthracyclines, its’ cumulative dose, and the length of follow-up as risk factors [15,16,17,18]. 

Survivors of childhood cancer compared to sibling controls are 15 times as likely to suffer from heart failure and are eight times more likely than the general population to die from the cardiovascular-related disease [19]. Our study confirms the cardiotoxic effects of anthracycline treatment. Two patients developed cardiomyopathy, one of them underwent heart transplantation and died from nonadherence to immunosuppressive treatment, the other is managed conservatively. A Japanese study on long-term complications in hepatoblastoma survivors also documented such observations. In their study, 17 out of 192 (8.8%) children had cardiac complications [4]. It has been reported that the risk of heart failure increases 11-fold in patients receiving doses over 300 mg/m^2^, as compared to ones treated with doses below 300 mg/m^2^ [20]. In our study, both patients with cardiac toxicity received a cumulative anthracycline dose of 360 mg/m^2^ or over. Since cardiotoxicity was observed in patients receiving much lower doses of doxorubicin, suggesting the role of individual susceptibility to anthracyclines, all patients treated with anthracyclines should have cardiological follow-up [21].

Second malignant neoplasms (SMN) are the most deleterious late effects of anti-cancer treatment [22]. In a nation-wide Korean study by Ju, out of 492 children with primary liver tumors, four (0.8%) developed SMNs. No information is available on the types of SMN [23]. While in the Hiyama study, out of 300 hepatoblastoma patients who survived over five years, SMNs occurred in 13 patients (4.3%). There were nine leukemias, one myelodysplastic disease, one lymphoma, one Ewing sarcoma, and one thyroid tumor [4]. In our group, 6.6% of patients developed SMN, but excluding patients with genetic predisposition who developed colon carcinoma, it drops to 4.4%. The remaining two children developed leukemia and medulloblastoma. There is a report on the website of the Children’s Hospital of Philadelphia on a patient who was treated successfully of hepatoblastoma at the age of nine months and six years later he developed medulloblastoma [24]. The case is similar to ours. The role of hepatoblastoma chemotherapy on the development of SMN is not obvious. Teepen et al., based on a study which included 6165 five-year childhood cancer survivors diagnosed between 1963 and 2001 in the Netherlands, strongly suggest that doxorubicin exposure increases the risk of subsequent solid cancers. [25] The same suggestions come from the Japanese study [4]. All of our patients who developed SMN were treated with doxorubicin. The role of the carcinogenic effect of chemotherapy for hepatoblastoma and SMN needs further studies. One may also speculate its role in early colon cancer development in patients with FAP mutation. 

Worthy of note is our patient who developed colon carcinoma prior to later diagnosis of FAP, which demonstrates different SMN etiology than in the two other patients. Trobaugh-Lotrario et al. identified 35 similar patients often in association with advanced colorectal carcinoma. The authors underline a need to identify patients earlier with germline adenomatous polyposis coli (APC) mutations for early colorectal carcinoma screening [26].

Hepatoblastoma may present with signs of precocious puberty, but no data is available on pubertal development of hepatoblastoma survivors [27,28]. None of our patients had signs of precocious puberty at diagnosis, and during further follow-up had normal stages of pubertal development.

In our study, we have found that more than half of our survivors had low BMI. There was no clear correlation between the children’s BMI at birth, at diagnosis, and at the time of follow-up. A study by Bruower has shown that anthracycline-treated survivors have more underweight at final height [29]. Nine of our underweight survivors had normal BMI at the time of diagnosis, which may support the role of chemotherapy on nutritional status. In our study, we have also observed that four transplanted patients were underweight, which could be attributed to both chemotherapy and post-transplantation treatment. More time is needed to evaluate our findings since none of our patients with low BMI reached their final height.

Bone mineral deficits have been reported after treatment for a variety of pediatric malignancies and are likely to be related to treatment with prolonged steroids and irradiation [30,31]. There is no specific information related to hepatoblastoma treatment and bone mineral deficits. 

It has been noticed that unexpected bone fractures can occur in patients with hepatoblastoma at diagnosis. Towbin et al. reports that out of 45 hepatoblastoma patients, eight (17.8%) had bone fractures [32]. He concludes that such findings are relatively common. In our series, three out of 45 patients during follow-up of more than five years were diagnosed with osteopenia, which resulted in fractures in two patients and idiopathic scoliosis in one. The pathophysiology of osteopenia in hepatoblastoma remains uncertain. 

Chronic kidney disease caused by cisplatin may affect more than 60% of children treated of neoplasm. Tubular damage leads to magnesia and chronic hypomagnesemia. Hypocalcemia may occur and is usually secondary to hypomagnesemia. This may occur in 10–30% of patients [33]. One of our patients developed tubular damage requiring supplementation of magnesium. Additionally, all patients treated with calcineurin inhibitors after liver transplantation need magnesium supplementation due to drug-related magnesia. One should also have in mind that significant number of pediatric liver transplant recipients develop chronic kidney disease as adults during long-term post-transplant follow-up due to immunosuppressive drug nephrotoxicity [34].

Other health adverse outcomes in our patients as epilepsy and nocturnal enuresis are unlikely to be treatment-related but add adversely to their final health status, especially that one-third of our hepatoblastoma patients experience at least two adverse health outcomes. 

In a subgroup of patients with unresectable tumors who underwent total hepatectomy and liver transplantation, long-term survival is about 80%. It is, however, achieved with the risk of additional surgical and immunosuppression related complications, which may influence patients’ health status in the long-term follow-up [35]. Among our liver recipients, late surgical complications prevailed and were the reason for liver transplantation in one patient. Longer follow-up is needed to evaluate long-term health outcomes in these patients. 

## 4. Materials and Methods 

All participants in this study were treated in our institution (Department of Oncology and Department of Pediatric Surgery and Organ Transplantation) and followed-up regularly in our pediatric oncology out-patients and late effect clinic. Patients who had received liver transplantation were also followed-up by the transplant team. Patients had to be at least five years from the diagnosis of hepatoblastoma, thus, the last patients included in our study were diagnosed in 2014. Finally, 45 patients were enrolled in the study. At the time of health evaluation, the youngest patient was 5.5 years old and the oldest 21 years of age (median—10 years). Time from diagnosis to last observation ranged from five to 22 years (median—nine years).

Data for this analysis were collected from medical records. Any missing information was complemented at the following visits. 

Recorded data included sex, gestational age, weight at birth and before treatment, age at hepatoblastoma diagnosis, disease stage, treatment methods (the type of surgery and chemotherapy), time since diagnosis, and evaluation of health status domains. Health status domains included performance status, growth development including weight, height, puberty, hearing, cardiovascular, skeletal, gastrointestinal, genitourinary, neurological, and hematological function. 

Performance status was evaluated using the Karnofsky/Lansky scale.

Age- and sex-specific height and body mass index (BMI) percentile classifications were calculated for each hepatoblastoma case. In children (age < 18 years), BMI reference values of underweight, overweight, and obesity, as already published [36,37,38], were used. The pubertal stage was evaluated by physical examination using Tanner criteria [39]. Absolute hearing levels were assessed and graded according to the Brock criteria [40] grade 0 < 40 dB at all frequencies—minimal hearing loss, grade 1 ≥ 40 dB at 8 kHz only—mild hearing loss, grade 2 ≥ 40 dB at 4 kHz, and above—moderate hearing loss, grade 3 ≥ 40 dB at 2 kHz and above—marked hearing loss, grade 4 ≥ 40 dB at 1 kHz—severe hearing loss. 

Most patients did not have baseline, pretreatment audiograms because they were too young or too sick. Of 45 patients, 39 (86.6%) had normal newborn hearing screening tests. 

Heart function was evaluated by electrocardiogram (ECG) and echocardiogram (ECHO), which were performed every 1–3 years. Cardiac toxicity was identified when symptoms of cardiac insufficiency were observed or in case of the presence of echocardiographic abnormalities; ejection fraction <50%, shortening fraction <28%, left ventricular posterior wall thickness z-score <2 or left ventricular end-diastolic dimension z-score >2. Shortening fraction and left ventricular ejection fraction (LVEF) were measured.

Skeletal system abnormalities were identified when symptomatic. Bone density was evaluated as a screening test in 12 patients using dual-energy X-ray absorptiometry (DEXA).

Other organ system toxicities were identified based on clinical or subclinical symptoms. Subclinical toxicities were defined as abnormal findings detected through blood tests specific for organ type (blood morphology, blood urea nitrogen (BUN), concentration of serum creatinine, glomerular filtration rate and creatinine clearance, liver tests, and on urine examination).

Academic achievements were evaluated by collecting information from patient/parent. At each visit, the patients/parents were asked about school performance, whether the children attended regular school, were in a class according to their age, needed extra lessons.

Second malignant neoplasms and deaths from circumstances other than hepatoblastoma were analyzed. 

The collected data were summarized with descriptive statistics.

Approval from the institutional Bioethical Committee for a retrospective medical record review was obtained (ethic code:12.06.2019). The database for this study is deposited on the institutional server under the supervision of the Director for Scientific Affairs of the Children’s Memorial Health Institute; e-mail: dyr.naukowy@ipczd.pl.

## 5. Conclusions

By a median of nine years from diagnosis, two out of three hepatoblastoma survivors had experienced at least one chronic health condition of any grade, which may have an impact on the quality of life (hearing loss) and late survival (SMN, cardiac failure). A growing number of children treated with liver transplantation will generate more adverse chronic health conditions in children with hepatoblastoma. 

Due to these morbidities, hepatoblastoma survivors should continue with regular long-term follow-up even after transmission to adult care.

Other adverse health conditions unrelated to hepatoblastoma treatment increase the burden of late effects in hepatoblastoma survivors.

## Figures and Tables

**Figure 1 cancers-11-01777-f001:**
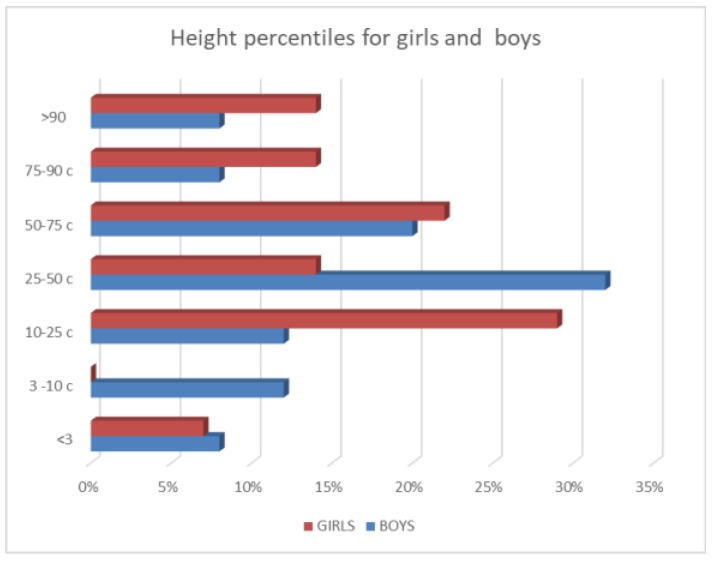
Distribution of percentiles for height.

**Figure 2 cancers-11-01777-f002:**
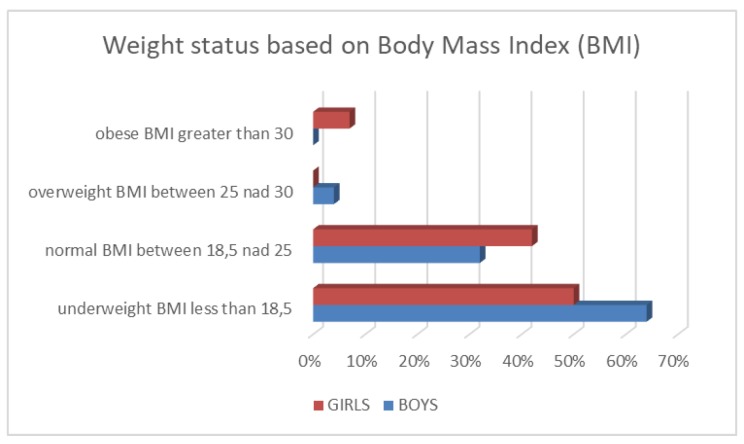
Distribution of percentiles for body mass index (BMI).

**Figure 3 cancers-11-01777-f003:**
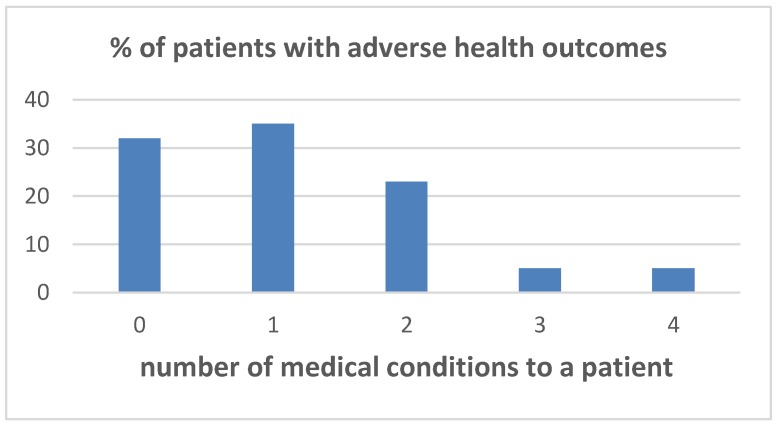
Adverse health outcomes.

**Table 1 cancers-11-01777-t001:** Characteristics of patients treated with liver transplantation (n = 9).

No/Male Female	Age at Diagnosis m-Months, y-Years	Age at Transplan-Tation	Indications for Transplantation	Age at Last Follow-up	Late Complications/Outcome	Other Late Effects
1.M	5 m	10 m	POSTEXT* III	10 y	Biliary complications—retransplantation two years later, 3rd transplantation within next 1 month due to primary poor function of the graft. During the next eight years several percutaneous transvascular balloon dilatations and stenting of the hepatic vein to inferior caval vein anastomosis were performed. Died of complications of idiopathic thrombocytopenia—10 years from diagnosis.	Brock 3 hearing impairment-hearing aids Acquired idiopathic thrombocytopenia
2.M	2 y	2 y 4 m	POSTEXT IV	16 y	None	None
3.F	7 m	1 y	POSTEXT III	13 y	None	Underweight BMI-16
4.F	1 y 4 m	1 y10 m	POSTEXT IV	14 y	Eight years from transplantation-hypersplenism due to portal vein thrombosis, treated successfully with partial splenic embolization.	Hypertension
5.M	14 y	14.5 y	POSTEXT IV	20 y	None	Hypertension, proteinuria
6.M	1 y 7 m	1 y 9 m		7 y	None	Underweight BMI-16
7.M	11 m	1 y 2 m	POSTEXT IV	6 y	None	None
8.M	3 y 10	4 y	POSTEXT IV	9 y	Late biliary anastomotic stenosis which was treated successfully with percutaneous transhepatic balloon dilatation	Underweight BMI 13, height <3 percentile Brock 3 hearing impairment-hearing aids
9.M	1 m	4 y	biliary cirrhosis four years from the primary resection of large volume tumor	6 y	1.5 years after transplantation-thrombocytopenia treated successfully with steroid pulses and rituximab. Single percutaneous balloon dilatation of portal vein anastomosis.	BMI 16, height <3 percentile

POSTEXT*—staging of tumor extent after neoadjuvant chemotherapy.

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
