# Peer review of "Health Status in Long-Term Survivors of Hepatoblastoma"

_cancers, 2019, doi:10.3390/cancers11111777_

Round 1

Reviewer 1 Report

This is a resubmitted paper about health status in follow-up of patients with hepatoblastoma treated at a single hospital. Unfortunately a document explaining the answers to each comment of the reviewers is not available. The highlighted sections may address these comments, but it remains unclear if all comments are sufficiently answered.

The questions of the reviewers are all addressed. Not all answers are
satisfying, but I seem no further improvement due to lack of info from
the patient. So I have no further comments.

Author Response

Dear Reviewer,

Thank you for the second round of remarks to our manuscript.

We don’t understand why you did not receive our answers to your comments. We tried to address all your issues, as you did not raise any new objections we are including again previous response.

Reviewer 1.

Comments and Suggestions for Authors

COMMENTS on paper entitled “ Health status in long term survivors of hepatoblastoma”, by Bożenna Dembowska-Bagińska et al.

The aim of the study was to evaluate health status of 45 children cured from hepatoblastoma. All patients were treated according to Childhood Liver Tumours Strategy Group - SIOPEL recommendations including cisplatin and doxorubicin.  Ototoxicity, second malignant neoplasms and cardiomyopathy were the most frequent late effects. They conclude that survivors of hepatoblastoma are at risk for long-term complications and require long-term monitoring for late effects even after transmission to the adult care.

The observation that children treated with cisplatin and doxorubicin are at risk for late effects  was described years ago. Unfortunately, This paper does not seem to add much to the bulk of knowledge on this topic.

Thank you very much for your thorough review of our paper and comments that you have formulated.

Your remarks were fully endorsed, more they contributed to refinement of our paper.

I will try to give answers and revise the manuscript according to your comments which hopefully will satisfy you.

Answers

 Yes, it is well known that children treated with cisplatin for different malignancies develop ototoxicity, same for anthracyclines and risk of cardiotoxicity.

Despite this already acknowledged facts on late effects there are scarce data which would assess health status (in general) of hepatoblastoma survivors. Even such a big group as SIOPEL did not publish data from their studies on late effects (in total). So, when one looks for second tumors, or other health conditions in hepatoblastoma survivors one can find data from series, usually concentrating on one health condition eg. ototoxicity.  This is one of the reasons we have undertaken this study. We are aiming at optimizing care for survivors of childhood cancers including ones with hepatoblastoma thus all available collected data providing information on our patients will facilitate multidisciplinary collaboration in patients care and early intervention. Our study hopefully might / will contribute to this issue.

Although the aim of the manuscript is interesting, the manuscript is lacking some fundamental data.

Authors should report more data on their patients and treatment: i.e.

a.       page 2, line 56. Physical development.  Data on weight and height pretreatment are not reported.

We have supplemented the data you have required (placed it in the manuscript). We have also added information on prematurity, weight at birth for all patients.

“Among 45 patients 7 (15.5%) were prematurely born. Their gestational age ranged from 28 to 36 weeks (median -34 weeks), weight ranged from 580 to 3080 grams (median – 2060 grams). Out of 7 premature babies, 1 was below 3rd percentile for weight the rest were over 50th percentile.  In the whole group 31 of 45 (68.8%) patients’ weight at birth was over 50th percentile while at hepatoblastoma diagnosis 18/45 (40%) had body weight over 50th percentile. At the time of treatment 15.5 % of patients (all boys) were below 10th percentile for height.

b.      page 3, line 69. Hearing. Children receiving cisplatin more than 400 mg/sqm are at higher risk for ototoxicity. An additional risk factor is the young age.

Yes, age is an additional risk factor. All but 2 of our patients were under 6 years of age, the other 2 were 8 and 14 years old

In this section, data on cisplatin schedule administration should be reported: how long was the infusion? Magnesium or thiosulfate was supplied? Mannitol was administered?

Cisplatin was administered as a 24 - hour infusion in all patients. Magnesium was supplemented during hydration. None of the patients received thiosulphate. Mannitol was administered according to chemotherapy protocol recommendations. We included the above in the “methods”

                c. page3, line 73. Cardiovascular status. More data on cardiological follow-up are needed: the followup was regular? Patients received treatment with enapril? Etc…

The cardiological follow-up was scheduled every 1 to 2 years. No patients received enapril, 1 patient received cardioprotection with dextrazoxane. We included it in “methods” section.

c.       page 3, line 81. Osteopenia. How was evaluated the osteopenia? Fractures were in syndromic patients?

Bone density was evaluated as a screening test in 12 patients using dual-energy X-ray absorptiometry (DEXA). All 12 patients had normal bone density. In 2 patients bone fractures were the first symptom of osteopenia. The fractures were the result of minor trauma.

                e. page 3, line 87. Nocturnal enuresis is not a common late effects of cisplatin and doxorubicin.

I agree that nocturnal enuresis is unlikely to be treatment associated but all adverse health episodes were recorded and reported.

Some reported late effects attributed to the antineoplastic treatment could be not really due to the chemotherapy( i.e., see epilepsy in Neurological disorders).

Yes, but again all health problems were recorded and reported as they influence the overall health status of childhood cancer survivor

Reviewer 2 Report

Author's done a good job to look at long term life quality after cancer treatment. All of them know about short time tumor treatment pain lke hair loos, loss of body weigh, many people not familiar with long term effect after the tumor treatment. Previously, several reports showed the toxicity of  cisplatin and doxorubicin like cardiotoxicity, nephro toxicity. Current study give clues to look at BMI and bone strength measurement. These two significantly impaired in the survivors of Hepatoblastoma treatment. The study needs more no of patients to get enough conclusion. 

Author Response

Dear Reviewer,

Thank you very much for your comments to our paper. We agree to your opinion that this kind of study needs more patients to draw final conclusions.

But in the literature thera are not many such reports addressing general healths status not only typical complications after tumor treatment. In this perspective we think that analysis of relatively big number of patients coming from one institution is of value in giving additional guidelines for the long term follow up.  Only multicenter studies can solve problem of small study groups in such rare disease.

Reviewer 3 Report

This paper describes the long term outcomes of patients treated for hepatoblastoma. The patients have been treated with standard surgical techniques and chemotherapy interventions.

The outcome measures described are well documented for the type of treatment that these patients have received. It is not clear from the paper what is novel here, or what new information the readership will gain from this study.

Author Response

Dear Reviewer,

Thank you very much for your comments to our paper. In the literature there are not many reports which address address general healths status in a holistic way rather than in direct relation to the hepatoblastoma treatment. We consider it very important for the patients especially in the transition from pediatric to adult care as  some problems will continue or even they can appear already in adulthood in our patients. We included in this long term assessment our first group of patients in whom liver transplantation was performed as an rescue surgical treatment, because this is another new patient with complex consequences of cancer, , chemotherapy, transplantation, immunosuppression and possible combined late general health status problems. 

Round 2

Reviewer 1 Report

The paper did improve after revision. Further improvements seem not to be possible. The paper iprovides a single center experience in late effects of children with hepatoblastoma. To generalize the conclusions larger number of patients are needed.

Reviewer 3 Report

My original review comments are unchanged by this revision